# A Visual Dashboard to Track Learning Analytics for Educational Cloud Computing

**DOI:** 10.3390/s19132952

**Published:** 2019-07-04

**Authors:** Diana M. Naranjo, José R. Prieto, Germán Moltó, Amanda Calatrava

**Affiliations:** Instituto de Instrumentación para Imagen Molecular (I3M), Centro Mixto CSIC—Universitat Politècnica de València, Camino de Vera s/n, 46022 Valencia, Spain

**Keywords:** visual learning analytics, learning analytics, learning dashboards, cloud computing

## Abstract

Cloud providers such as Amazon Web Services (AWS) stand out as useful platforms to teach distributed computing concepts as well as the development of Cloud-native scalable application architectures on real-world infrastructures. Instructors can benefit from high-level tools to track the progress of students during their learning paths on the Cloud, and this information can be disclosed via educational dashboards for students to understand their progress through the practical activities. To this aim, this paper introduces CloudTrail-Tracker, an open-source platform to obtain enhanced usage analytics from a shared AWS account. The tool provides the instructor with a visual dashboard that depicts the aggregated usage of resources by all the students during a certain time frame and the specific use of AWS for a specific student. To facilitate self-regulation of students, the dashboard also depicts the percentage of progress for each lab session and the pending actions by the student. The dashboard has been integrated in four Cloud subjects that use different learning methodologies (from face-to-face to online learning) and the students positively highlight the usefulness of the tool for Cloud instruction in AWS. This automated procurement of evidences of student activity on the Cloud results in close to real-time learning analytics useful both for semi-automated assessment and student self-awareness of their own training progress.

## 1. Introduction

The last years have witnessed unprecedented advances in the education field with the rise of on-line education platforms and highly successful MOOCs (Massive Online Open Courses). These courses are powered by the technological advances in multimedia production and the widespread presence of high bandwidth communication networks across the globe. Indeed, there is a common trend in students wanting to learn anywhere and anytime without the inherent barriers of traditional face-to-face education [1]. This has paved the way for new educational approaches to surge such as blended learning [2], which combines online multimedia material with traditional face-to-face classroom, or flipped learning [3], a pedagogical approach in which instruction shifts away from the classroom into individual learning and the classroom is used as an interactive learning environment [4,5].

Computer Science and Computer Engineering degrees in higher education institutions have also embraced this change and are starting to adopt techniques to foster out-of-class activities. In the field of distributed computing there have been previous experiences by the authors adopting Cloud computing to support the management of online courses [6] and to deploy highly available massively scalable remote computational labs [7]. As described in the work by Gonzalez et al. [8], many higher education institutions are adopting Cloud computing to benefit from reduced maintenance costs, rationalization of resources and simplified operation. For this, public Cloud providers such as Amazon Web Services (AWS) provide the required hardware infrastructure on which to carry out hands-on lab sessions for different distributed computing subjects.

There exists the AWS Educate program (AWS Educate: https://aws.amazon.com/education/awseducate/) by which academic institutions, professors and students can apply for credits to offset the charges resulting from resource consumption. However, the prerequisite of setting the student’s credit card details to access the whole set of AWS services instead of a restricted environment (Starter Account) is a serious limit to this approach. Therefore, an effective approach consists of having a master AWS account owned by the instructor and shared by the students by means of specific user accounts linked to the master account with restricted privileges [6].

Under these circumstances, the instructor requires insights on the way that students are using the different AWS services to account for excessive usage and to be able to distinguish among the students that are carrying out the proposed activities from those that are not. Fortunately, the AWS CloudTrail [9] service enables compliance together with operational and risk auditing of an AWS account so that the activity that occurs within is recorded in a CloudTrail event through a set of virtual sensors distributed across the supported services which are automatically managed by AWS. However, the dashboard offered by CloudTrail has serious limitations. On the one hand, it does not allow to perform complex queries involving several search parameters, something that it is required for the instructor to identify the activities carried out by the students. On the other hand, it restricts queries of events to a maximum of 90 days, which is clearly insufficient to gather information that spans a whole academic year. Finally, the CloudTrail dashboard has not been designed with an educational-oriented goal and does not allow to easily show the progress of students with respect to the hands-on lab activities.

The amount of data that is available for analysis in this scenario is increasing considerably. New research areas have emerged in order to take advantage of these data to improve the learning process of students and teachers. Thus, *Learning analytics* has been defined as *“the measurement, collection, analysis and reporting of data about learners and their contexts, for purposes of understanding and optimizing learning and the environments in which it occurs”* [10]. This concept is strongly related with *learning dashboards*, defined as single displays that aggregate different indicators about learner(s), learning process(es) and/or learning context(s) into one or multiple visualizations [11]. The fundamental idea behind these concepts is to allow users to track their activities, in order to achieve self-analysis and comparison with other users, motivating users to perform the proposed activities and improving self-regulated learning by visualizing these activity traces and what are the activities pending to be carried out. As Sedrakyan et al. [12] state, effective feedback needs to be grounded in the regulatory mechanisms underlying learning processes and an awareness of the learner’s learning goals.

To this aim, this paper introduces CloudTrail-Tracker (https://www.grycap.upv.es/cloudtrail-tracker), an open-source serverless platform for enhanced insights from CloudTrail logs for multi-tenant AWS accounts. This has been particularly tailored for the educational field in order to provide a web-based blended dashboard that offers the instructor aggregated information on the usage of the AWS services by the students and detailed usage information of AWS by a specific student on a given time frame, together with a percentage report of their fulfilment of the different lab sessions. This information is offered to the student through a customized visual dashboard in order to foster self-regulation by indicating the progress and the pending actions for each activity lab in educational activities for AWS trainining. We aim to deliver process-oriented feedback when carrying out hands-on labs that can help teachers and learners foster engagement and achievement. To the best of the authors’ knowledge this is the first learning dashboard for Amazon Web Services, freely provided as an open-source development for the benefit of the academic community.

After the introduction, the remainder of the paper is structured as follows. First, Section 2 introduces the related work in the area of learning analytics dashboards. Next, Section 3 describes the application architecture and briefly provides additional technical details. Later, Section 4 describes the subjects and courses in which the tool is being used to provide automated compilation of evidences of the work carried out by students. Then, Section 5 discusses the benefits and the possibilities introduced by this tool. Finally, Section 6 summarizes the main achievements of the paper and points to future work.

## 2. Related Work

Learning analytics is a topic that has gained relevance in the last years with the rise of automated data collection and data processing techniques, together with the surge of MOOCs. As an example, Tabaa et al. [13] designed a learning analytics system that deals with the huge amounts of data generated by MOOC platforms, whose main aim is to automatically detect students at risk of abandoning the studies. A recent review in this field can be found in the work by Patwa et al. [14], that outlines the importance of learning analytics, the current resources, and the challenges that it presents.

Learning analytics has paved the way for learning dashboards to appear in order to provide a visual interpretation on the progress of students. For example, the work by Schwendimann et al. [11] reviews the state-of-the-art regarding research trends on learning dashboards. They propose a definition for *learning dashboards* and point out the main needs of the field, which lacks from validation, comparison between solutions and aggregated data from different fields. The work by Verbert et al. [15] presents a review about dashboard applications to support learners and teachers in on-line environments and also in classroom environments. It also analyzes the main challenges to address, such as the deployment and configuration of the dashboards and the choice of sensors used to collect the data.

As it is reflected in the reviews, several tools have appeared in the very last years. Remarkable tools in the field are Course Signals [16], a dashboard that predicts and visualizes learning outcomes based on grades in the course, time on task, and past performance, and Student Activity Meter (SAM) [17], a dashboard that provides visualizations of progress in online courses for teachers and learners, focusing on the awareness of time spent and resource use. Other tools available in the literature are VisCa [18], a web-based dashboard system to track, store, and show learning status from e-learning platforms; LOCO-Analyst [19], a tool for teachers to analyze the performance of their students, and GLASS (Gradient’s Learning Analytics System) [20], a web-based visualization platform based on modules that provide different configurable visualizations derived from a common dataset.

There also exist dashboards specifically designed for mobile devices, such as StepUp! [21], a mobile app for the students that applies learning analytics techniques for awareness and self-reflection. The work by Vieira et al. [22] is the most recent analysis in the field of visual learning analytics. The authors state that there is a lack of studies that both employ sophisticated visualizations and engage deeply with educational theories, a statement also supported by Jivet et al. [23], where learning dashboards are analyzed from the point of view of learners. However, although there are several learning dashboards in the literature, none of them tackles the field of Cloud Computing studies.

Focusing on Cloud platforms, Amazon Web Services offer two solutions to monitor the usage of its resources, AWS CloudTrail [9], a managed service to track user activity and API usage, and Amazon CloudWatch [24], a monitoring service of Cloud resources and applications. However, none of them are sufficient for monitoring AWS resources and services when applied to an educational context. In particular, the oldest event that can be queried in CloudTrail is 90 days, well under the span of an academic year. Moreover, a learning dashboard requires more advanced analytics such as aggregated usage across a period of time. Indeed, there are several alternative solutions in the market that offer more powerful dashboards for monitoring Cloud resources. Some of them are Spectrum [25], Opsview Monitor [26], SignalFx [27] and AWS Cloud Monitoring [28]. However, all of them are costly enterprise solutions that are beyond the reach of academic institutions.

Recent research on the effectiveness of learning analytics tools highlights that using performance-oriented dashboards might decrease learner mastery orientation and that students’ exposure to graphics of their academic performance may negatively affects students’ interpretations of their own data as well as their subsequent academic success, as described in the work by Lonn et al. [29]. Therefore, our goal is to provide fast feedback on the activities carried out by the students during the hands-on labs for reinforced feedback, rather than focusing on academic performance. This is inline with the work by Sedrakyan et al. [12] where students are provided with process-oriented feedback aimed at having an impact on their behaviour. The ability to provide timely feedback is crucial because, as identified by Irons et al. [30]: the sooner the feedback is delivered to students, the more impactful it is for their learning.

From the analysis of the state-of-the-art we can extract that the big challenge is how data coming from the learning process can be meaningful for different profiles, such as teachers and students, when using Cloud computing. To this aim, the main contribution to the state-of-the-art of this paper is a blended learning dashboard that combines information concerning the usage of resources in a shared AWS account by multiple students together with the degree of progress of the students with respect the hands-on lab activities to be carried out, accesible both for the instructors and for the students.

## 3. Materials and Methods

This section describes CloudTrail-Tracker and provides insights on the underlying technology employed for its development to create the blended learning dashboard for AWS. First, the back-end of the application will be described, in order to store the actions carried out by the students in AWS. Second, the educational dashboard will be addressed so that both students and teachers can have an overview of the activities performed in the Cloud.

### 3.1. Architecture

The architecture of the application is shown in Figure 1. The flow of data starts with the students using any of the AWS services involved in the educational activities. The CloudTrail service registers and retains a certain window of the history of events related to the activity of an AWS account. It records the actions performed by the students as a set of events that describe who used what and when. These data are stored as a set of files in Amazon S3 (https://aws.amazon.com/s3), an object storage service that uses *buckets*, a container used to store the files. Amazon S3 stores large amounts of data that can be retrieved from anywhere at any time through a web services interface.

A sensor is considered a device, module, or subsystem whose purpose is to detect events or changes in its environment and send the information to other electronics. In this regard, AWS CloudTrail mainly tracks invocations done to the APIs of the different AWS services, which are typically performed by either the AWS Management Console or the AWS CLI (Command Line Interface), on behalf of the user, in order to create a centralized log of the actions performed by the users of an AWS account. Therefore, CloudTrail acts as virtual sensor that traces the activity of the users in an AWS account and sends the information to Amazon S3. These data, when properly processed, can be used to map the activities carried out by the user into the steps of a learning activity in the Cloud.

Whenever a new file is created in the bucket, an AWS Lambda (https://aws.amazon.com/lambda) function is triggered that parses this file and stores the relevant fields into Amazon DynamoDB (https://aws.amazon.com/dynamodb). AWS Lambda supports creating functions triggered by events without the need of explicit management of servers. This is commonly called a *serverless* application, in which AWS Lambda executes code with automated scaling and high availability featuring a fine-grained pay-per-use pricing model where no costs are incurred if the function is not being invoked. Amazon DynamoDB is a scalable, high-performance, fully managed database service that enables the storage of key value pairs with very low data access latencies and optimal scalability.

In order to be able to query for these events, an API (Application Programming Interface) is created in API Gateway (https://aws.amazon.com/api-gateway) that, upon every request, triggers the execution of a Lambda function that queries the events in DynamoDB. API Gateway is a fully managed service that allows developers to create, maintain, monitor and protect APIs at any scale. API Gateway is usually integrated with Lambda so that a request to the API triggers the execution of a Lambda function to process that invocation. This allows creating a service in the Cloud whose economic cost only occurs when it is used.

Therefore, CloudTrail-Tracker is implemented as an event-driven *serverless* application that involves no Virtual Machines in the cloud. Depending on the number of students (in the order of hundreds) it can operate within the AWS Free Tier (https://aws.amazon.com/free/), thus collecting, storing and serving the events at zero cost. AWS provides the underlying software services required to efficiently and cost-effectively operate this application in production regardless of the number of students. However, apart from CloudTrail, in charge of producing the user activity logs, any other component in the architecture could be replaced by an open-source alternative or it could use the corresponding software service from another major public Cloud provider, such as Microsoft Azure or Google Cloud Platform.

To facilitate the access to these information, a web-based application has been created that queries API Gateway to produce high-level aggregated information both for students and teachers. This is the basis of the educational dashboard.

### 3.2. Dashboard

Analytical educational dashboards provide teachers with various information about the skills, the progress, the performance and the mistakes made by students [31]. The use of educational dashboards can quickly and efficiently transform the information related to the details of the learning process and provide the students and the instructor with the necessary information to track its evolution. A well-designed dashboard makes the data easier to understand and its presentation in an interactive environment between the student and the teacher leads to interesting debates [32]. However, there are no golden rules for the design of educational dashboards, since this depends on the requirements to be achieved and the information to be shown [33]. We aim to influence on negative emotions such as lack of interest and the perception of being lost, which are detrimental to student learning [34], by means of guided indications for students to achieve the goals set in the practice labs.

Therefore, in order to achieve a high-level overview of the activity of the students carried out in AWS, we designed a Learning Dashboard that provides:Aggregated information concerning the usage of resources in AWS in a certain time frame specified by the user.Detailed information concerning the specific activities carried out by a certain student in a certain time frame.Percentage of progress for a certain student with respect to all the hands-on lab activities, defined by the instructor, carried out in a certain time frame.

For the authentication of the users and the protection of the API, we use Amazon Cognito (https://aws.amazon.com/es/cognito/), a service that provides access control, registration and login for users in web or mobile applications. When a user authenticates to the application, the access credentials (username and password) are sent to Amazon Cognito, where the credentials are authenticated and an access token is obtained to communicate with API Gateway.

For the development of the front-end, Vue.js was used because it supports creating user interfaces with intuitive, modern and easy-to-use features, it has a very active community and it is very easy to integrate with existing applications. In addition, using this framework allows to easily generate a static website (HTML + CSS + JavaScript) so that it can be served from an Amazon S3 bucket, thus offering a scalable and very low cost access to the application’s web panel.

One of the most important issues to take into account in the development of any web application is the ability to access it from any device (mobile, tablet, laptop). In order to make the application accessible from any device, a responsive theme was used that adapts to any screen size and offers an improved user experience across multiple platforms.

The users of the educational dashboard fall into one of these different roles:**Teacher**. Users with this role want to visualize the progress of each student in the course, and also for each lab session. Useful metrics to guide the assessment of the labs are shown, such as the degree of completion for each lab session. The benefits of the dashboard for this role are to monitor multiple students at a glance, providing automated feedback for the learner and obtaining automated metrics that may be used for assessment.**Students**. Users with this role expect from the dashboard to track their progress of each lab session, become aware of what activities are missing to complete their tasks and, only if the instructor is in favour of this, compare their progress with the rest of the class. The benefits of the dashboard for this profile are: self-regulation learning, including planning, judgments and evaluation of tasks and context [35], motivation [36] and a general view of the class.**System Administrator**. The dashboard benefits the users with this role by facilitating the view of the aggregated and detailed resource consumption, including the visualization of historic usage of the resources, and monitor the current use of resources (close to real-time, since there is a 15 min delay from the account activity until the creation of the log in CloudTrail). Thus, the administrator can detect irregularities in the usage of the resources in order to avoid overspending.

Figure 2 shows the aspect of the dashboard for the teacher. It provides a summary of the most important resources, i.e., those with higher cost, provisioned by students in AWS. This allows the teacher, or the administrator, to detect abnormal behavior such as spikes in resource consumptions caused by leaked access credentials to the Internet. Notice that this information aggregates resource usage across the multiple regions offered in AWS, a feature that can only be achieved with the AWS Management Console by switching from region to region, thus simplifying the job for the teacher and the administrator. A bar graph at the bottom of the page (not shown in the picture) describes the students that have been using the platform in the time period selected together with the number of actions carried out by each one. The dashboard allows to filter by specific course, since different activities, or in different order, may be carried out in each course.

In the case of synchronous training activities, such as face-to-face sessions, this panel allows to see in a glance the students that are lagging behind their peers. This represents an opportunity for an early intervention by the instructor to reinforce the students and provide additional support if required, especially for students that lack the self-confidence required to proactive seek for this support. For online instruction, the dashboard is also useful to identify the preferred time slots for students to carry out the lab activities, since self-paced instruction is offered on a worldwide scale. This may help identify and prevent soft limits reached due to the concurrent usage of resources in a shared AWS account by multiple students.

## 4. Case Study Results: Usage in Cloud Computing Education

CloudTrail-Tracker was developed during the academic course 2017/2018 together with a set of pilot experiences in several subjects and was rolled out in production for the academic course 2018/2019. Different training initiatives are benefiting from the ability to automatically track the activity of students in the Cloud, as shown in Figure 3. This involves three subjects across the same number of Master’s Degrees in which different approaches for learning are employed that range from face-to-face instruction to flip learning. The tool is also being used in the Online Course in Cloud computing with Amazon Web Services (http://www.grycap.upv.es/cursocloudaws).

In those scenarios, students are guided to use AWS services for the development of Cloud applications so that they can train the appropriate skills by doing a set of hands-on labs that showcase the main functionality of the AWS services involved. Therefore, a hands-on lab is defined as a set of ordered events (i.e., an action on an AWS service, such as creating a Load Balancer or deploying a Virtual Machine) that the students have to perform in order to consider the lab completed.

To detect the progress percentage of a student across the hands-on labs we created specific web panels in the educational dashboard responsible to compute whether a set of events, which defines a lab activity, is included in the set of events related to a student in a given time period. This is shown, as an example, in Figure 4, where the percentage of completion for each hands-on labs for a specific student is shown. By using a graph bar combined with a traffic light rating system, the teacher can see in a glance whether the student is progressing accordingly.

It is agreed that student’s learning improves and their understanding deepens when they are given timely and targeted feedback on their work, according to the work by Butler and Winne [37], in which they establish the link between feedback and self-regulated learning. Also, we aim at increasing effective action from the feedback, as described in the work by de Freitas et al. [38], in which they use gamified dashboards and learning analytics to provide immediate student feedback and performance tracking in Higher Education studies. Therefore, we provide students with access to the educational dashboard which is used as a tool so that students discriminate between that parts of the hands-on labs that are already done and those that are still pending.

It is important to point out that we allow students to become aware not only about the percentage of completion but also about the missing actions that are pending to be done, as shown in Figure 5. By including the missing actions for a specific student in each hands-on lab we achieve a two-fold objective. On the one hand, students become aware of the missing actions and they are provided with a chance to self-regulate and complete the activities. On the other hand, since students tend to forget about terminating and deleting the unused resources, which provoke an increase in the economic cost of resources and also represents a bad practice, we anticipate that this will result in a reduction of the economic cost.

The liaison between learning analytics and pedagogy is fundamental since they are both bound to epistemology, that is, the theory of knowledge. We designed our educational dashboard following a constructivist approach, as described in the work by Knight et al. [39]. Constructivist models focus on the forms of learning that happen during the learner’s guided exploration and experimentation with the environment, in our case, through the exploration and usage of the different Cloud services. Learning analytics based on constructivism approaches focus on progress, particularly through tracking and judging the modifications made to a set of resources arranged by the educator [39]. This is precisely the approach taken for the development of the CloudTrail-Tracker dashboard, which presents the information on bar diagrams for students to easily identify their progress using a traffic-light coloring system. This information is supplemented with specific missing actions to provide timely guided feedback.

With the help of the dashboard, the students have an overall perspective of their progress across all the lab activities to be performed in the course. They are particularly keen on watching the bars rise and turn to green while they complete the Cloud activities in AWS. Even if they have the lab guides for further support during the practical activities, the visual tracking of the progress allows them to clearly discern the progress being carried out.

The assessment strategies for these subjects involve assigning a certain percentage of the final mark to the completion of the hands-on labs together with auto-graded questionnaires that include questions concerning the main functionality of the services used and, depending on the subject, an academic work that integrates multiple services to create a real Cloud application. Therefore, having an overview of the activity of the student paves the way for semi-automated assessment of the work carried out during the hands-on labs.

### 4.1. Satisfaction Survey and Usage Analytics

Students from the different Cloud courses shown in Figure 3 were asked to voluntarily fill in an online survey using Google Forms in order to determine the degree of satisfaction with CloudTrail-Tracker. A 10-item Likert scale questionnaire was employed where 0 means strongly disagree and 10 means strongly agree. The background of the students across the subjects is diverse but mostly coming from technical studies (STEM). The results, for a population of 64 responses, are shown in Table 1. Table 2 provides a disaggregation of the results across the main educational activities carried out in which CloudTrail-Tracker has been employed (see Figure 3 to identify the subject from the acronym).

There were no significant differences between face-to-face students and fully online students. The results indicate that students perceived a high degree of usefulness of CloudTrail-Tracker as a tool to support educational activities on AWS. The ability to provide timely feedback on their lab progress by means of easy-to-use accesible web interfaces that provide convenient information was highlighted as a remarkable contribution. However, students indicated that there was room for improvement. They performed minor suggestions regarding the usability of the tool in order to obtain the list of pending events when clicking each bar, which has already been implemented, and pointed out that a better matching between the missing action and the specific part of the lab guide should be included. We are partially addressing this issue by including links to the official AWS documentation which properly explain each action. It is important to point out that these improvement suggestions do not affect the design of the underlying system but only require changes in the graphical user interface to better align the information provided by the tool with the expectations from the students regarding the amount of information for guidance that it delivers.

The tool was initially released in October 2018 and usage analytics started being tracked in February 2019. Concerning the usage of the dashboard, Figure 6 shows the statistics offered by Google Analytics restricted to a period in which both SEN (ending in mid-April) and CursoCloudAWS were taking place. The graph shows the number of users and the average session duration (the time spent using the tool) across the aforementioned period. The spikes correspond to days on which SEN sessions take place, where peaks of almost 15 students are achieved. Other relevant accesses come from either the instructor or students taking the self-paced online course. The average session duration appears to decrease along time, which might be an indication that students dedicate less time to tinker with the tool since the information shown can be rapidly understood after using it several times.

### 4.2. Data-Driven Course Reshape from Insights

The development of tools that provide further data-driven insights of the evolution of the students paves the way for course reshaping taking into account the behaviour of the learners. To this aim, this section provides an statistical analysis of the data obtained from the Online Course in Cloud Computing with Amazon Web Services, involving the aggregated information obtained by CloudTrail-Tracker concerning the progress of the students through the lab activities. It is important to point out that the study involves the use of historic data conveniently tracked by the AWS CloudTrail service. The use of CloudTrail-Tracker provided the ability to extract valuable information from the data regarding how the students behaved during the practice activities.

The study involves a population of 323 students that took the self-paced course during the academic years 2016/2017 and 2017/2018. The students have access to all the course material (video-lessons, lab guides, AWS access and self-assessment questionnaires) since the first day of enrolment and they are only required to pass a final test with 50 multiple-choice questions with varying difficulty that involves both theoretical and technical concepts from the labs, as it happens in typical computer-based certification exams. There is further information about this course, and how it was efficiently implemented using Cloud services, in the work by Moltó et al. [6].

Table 3 shows the average percentage of progress of the students for each lab activity carried out in AWS. They are shown in chronological order of appearance in the course material (i.e., PL_EC2 is the first one and PL_SERVERLESS is the last one). The results indicate that, on average, fewer number of students reach the end of the course performing all the lab activities proposed. This may be a symptom that further student engagement should be reinforced in the course together with an assessment of whether the time allocated to perform the activities for the course is properly dimensioned.

A histogram of the variable PL_TOTAL, which stands for the percentage of progress across all the practical activities for a given student is shown in Figure 7a. Notice that 72 people out of a population of 323 never carried out the lab activities. Indeed, online students are not forced to carry them out even though they are encouraged to do so in order to facilitate the understanding of the technical capabilities of each AWS service. These data, which could only be obtained through CloudTrail-Tracker, unveiled that the assessment of the course should properly reinforce the realization of the lab activities. The histogram of the final grades of the course, shown in Figure 7b show a bias towards good marks which indicates that stronger emphasis should be made in the assessment of the lab activities in the final test to better discriminate among students who carried out the lab activities and those who did not.

## 5. Discussion

Cloud computing provides the ability to offer scalable computational resources for students to carry out hands-on sessions. The ability to automatically track the usage of the AWS platform by each student allows the teacher to have an educational dashboard that provides an overview of the work performed by the student on each lab session. In addition, by disclosing these data to the students we aim to foster self-regulation, since students become aware not only of the activities already carried out but also of those still pending. It is precisely in that awareness that lies the ability of a student to master critical soft skills such as time management and planning. However, as indicated in the work by Jivet et al. [40], being aware does not imply that remedial actions will be adopted by the student and, what is more important, that the learning outcomes are improved. To this aim, the dashboard not only provides the ability to track the student progress, but also indicates the missing actions to be carried out.

It is important for students of computer science and computer engineering to understand public Cloud providers and, mostly, how to combine the multiple services offered to create applications that involve computation and data management applied to their own specific domains. However, this involves an economic cost that should be taken into account by students when designing an application architecture. Therefore, having a high-level graphical user interface that helps them know whether they properly terminated the allocated computing and storage resources reinforces this skill. As Arnold et al. [41] state: *“automating and scaling tools that can aid students in their monitoring (awareness), feedback, and adapting self-regulation practices is a complicated process”*. This is why we believe that Cloud computing provides a foundation platform to automate tracking the activities of students and, the development of a blended learning dashboard that provides both analytics for teachers, students and administrators paves the way for this awareness in the students to happen.

This functionality has to be encompassed by a system that is both cost-effective and runs proactively in the background collecting the evidences of the student activities anytime of the day through the virtual sensors automatically provided by AWS CloudTrail. While face-to-face instruction occurs at specific time instants, the case of the online course is different, since it is a self-paced training activity. This requires that the system in charge of collecting the evidences is up and running at all times. By designing an event-driven serverless architecture that operates within the Free Tier we achieve minimal cost (even zero cost depending on the level of use by the students) while ensuring that is fully operational around the clock.

The benefits of CloudTrail-Tracker with respect to the generic monitoring solutions for AWS described in the related work section lie in its ability to operate within the boundaries of the AWS account, without requiring to provide third-party services with access to the CloudTrail data. Also, the ability to define a lab session in terms of the corresponding matching events of the different AWS services involved is a unique feature of the platform. This allows for unobtrusive monitoring of the students’ activities when learning to interact with the different AWS services. Gathering these analytics and exposing them via easy-to-use web-based graphical dashboards allows students to discriminate between the activities carried out and those pending to be done. Finally, providing timely feedback to students with indications of the missing actions fosters self-regulation and the ability to achieve best practices for managing computational resources in AWS, such as terminating resources after finishing a lab session.

To the author’s knowledge, there is no educational dashboard referenced in the literature that provides automated compilation of the student activity in AWS in a high-level educational dashboard. CloudTrail-Tracker has been released as an open-source development available in GitHub (CloudTrail-Tracker GitHub repository: https://github.com/grycap/cloudtrail-tracker) so that other teachers can adopt it and report feedback.

## 6. Conclusions

This paper has introduced CloudTrail-Tracker, a platform that provides usage insights of an AWS account, which has been used to provide automated gathering of evidences of the activities performed by students on a shared AWS account. This has been possible by processing the data gathered by the virtual sensors distributed across the supported AWS services that collect usage data of the platform and centralize it through the AWS CloudTrail service.

The system can run at barely zero cost on an AWS account and it includes both an event processing back-end and a web-based educational dashboard that provides teachers with further knowledge on the way students are using AWS to carry out the activities proposed. This educational dashboard has been customized to support several subjects across three Master’s Degrees and an online course on AWS so that the teachers, and the students themselves, precisely know the degree of completion of each hands-on lab together with the actions missing with the aim of fostering student self-regulation. Also, the dashboard helps system administrators to detect irregularities in the usage of the resources.

The satisfaction results across a population of 64 students indicate that more than 90% of students are highly satisfied with the accessibility and the ease of use of CloudTrail-Tracker together with the facility to understand the information shown by the tool. They consider it an appropriate support tool for the education in AWS technologies, while there is room to improve the detail of information related to the progress of students for each lab activity.

Future work involves further customization of the information shown. For the teachers, we will include additional panels that show real-time monitoring across multiple services and regions in order to overcome the 15 min delay, to anticipate problems of excessive resource usage during a lab session. For the students, we plan to provide not only access to their history of events and degree of completion for each hands-on session, but also the average performance of their peers in the subject. For synchronous teaching activities we expect this to be a booster for students that tend to procrastinate. We also aim to provide additional detailed information concerning the missing actions for students to better understand what actions to be done next. Finally, we plan to introduce gamification techniques in the dashboard so that students can challenge themselves and other students to increase motivation when carrying out the activities.

## Figures and Tables

**Figure 1 sensors-19-02952-f001:**
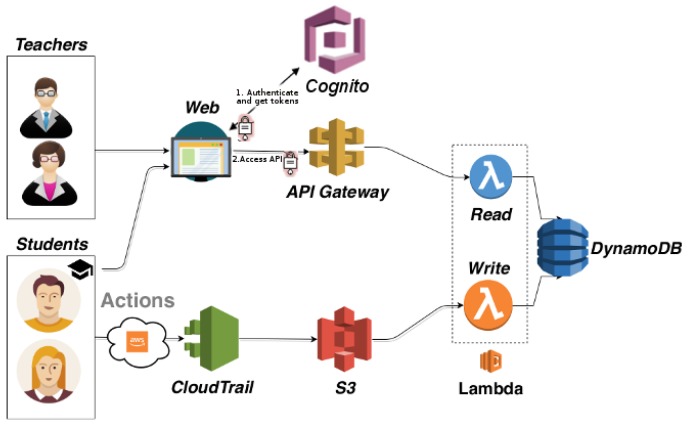
Architecture of CloudTrail-Tracker.

**Figure 2 sensors-19-02952-f002:**
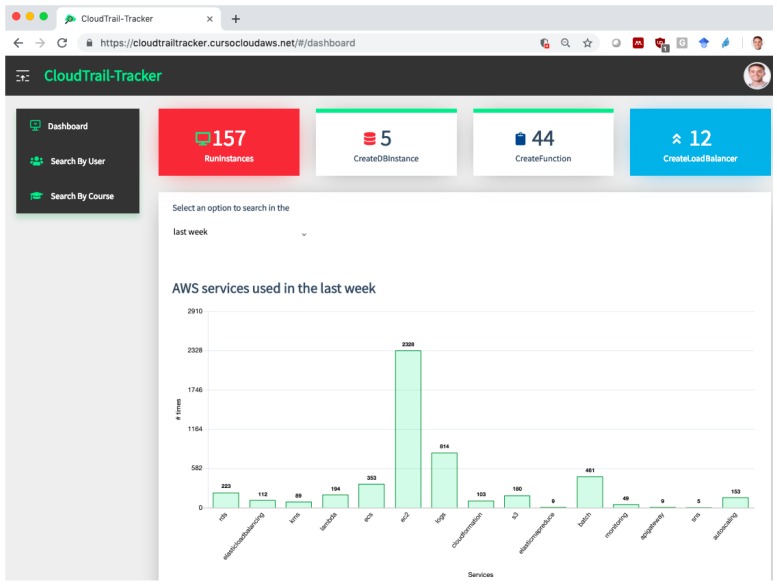
Initial dashboard provided by CloudTrail-Tracker showing aggregated usage statistics of the AWS account.

**Figure 3 sensors-19-02952-f003:**
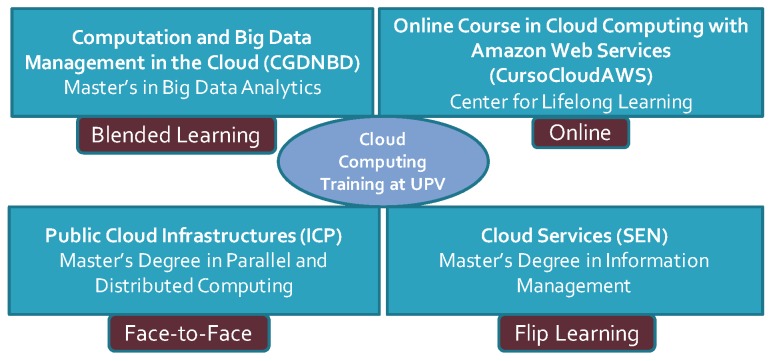
Cloud computing training at the Universitat Politècnica de València, in Spain.

**Figure 4 sensors-19-02952-f004:**
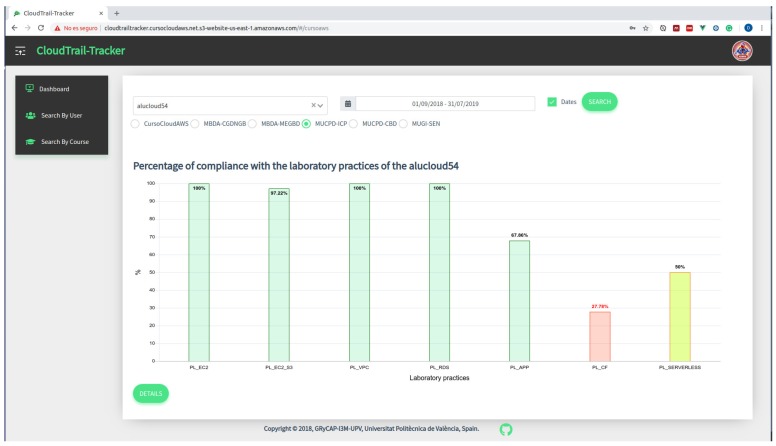
Percentage of completion for each hands-on lab for a specific student.

**Figure 5 sensors-19-02952-f005:**
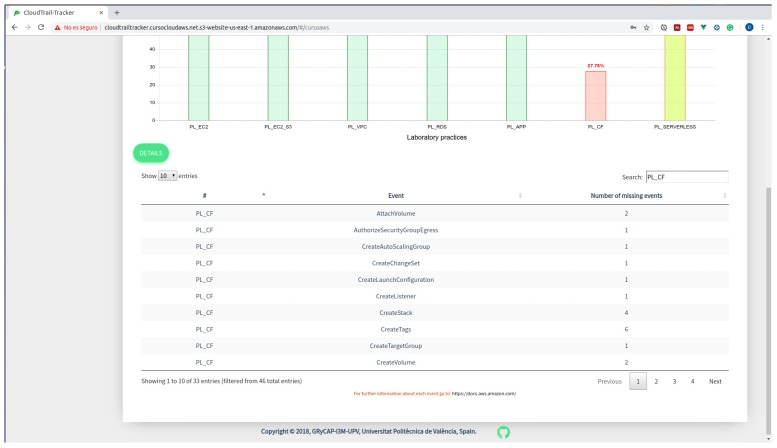
Missing actions for each hands-on lab for a specific student.

**Figure 6 sensors-19-02952-f006:**
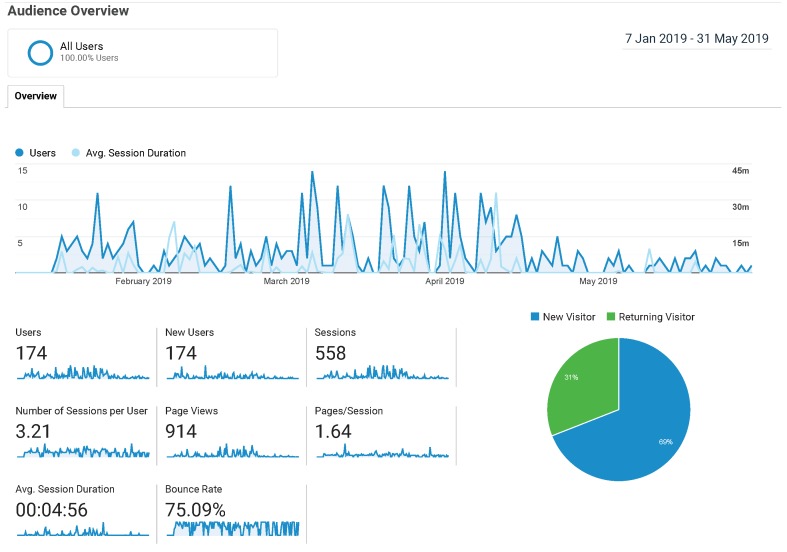
Usage analytics of the CloudTrail-Tracker dashboard.

**Figure 7 sensors-19-02952-f007:**
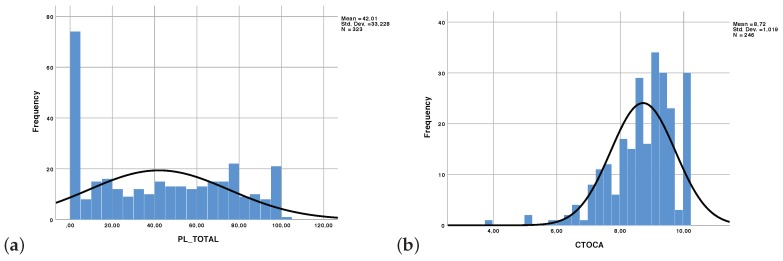
Histogram of (**a**) frequencies of the progress across all the lab activities (PL_TOTAL, for 323 students) and (**b**) final grade on a [0, 10] scale (CTOCA, for the 246 students who actually carried out the final test).

**Table 1 sensors-19-02952-t001:** Results of the satisfaction questionnaire with CloudTrail-Tracker (the percentage of students that answered in each interval, using a 10-item Likert scale, is shown).

Question	[0, 4]	[5, 7]	[8, 10]
Q1. The tool was always accessible whenever I needed it	0	3.1	96.9
Q2. I knew how to use the tool without the instructor’s guidance	0	6.3	93.8
Q3. I was able to properly understand the information given by the tool	0	7.8	92.2
Q4. The information shown by the tool helped me identify my progress in each lab	2.2	10.9	87.5
Q5. It can be considered an appropriate support tool for the education in AWS	0	9.4	90.6

**Table 2 sensors-19-02952-t002:** Results of the satisfaction questionnaire with CloudTrail-Tracker disaggregated by subject (*N* stands for the number of students that filled in the questionnaire for each subject, AVG stands for average and STD stands for standard deviation).

	CursoCloudAWS (*N* = 11)		SEN (*N* = 17)		ICP (*N* = 7)		CGDNBD (*N* = 28)	
	*AVG*	*STD*	*AVG*	*STD*	*AVG*	*STD*	*AVG*	*STD*
Q1	9.25	1.22	10.00	0.00	9.71	0.76	9.89	0.31
Q2	9.42	1.08	9.65	0.86	9.71	0.49	9.00	1.22
Q3	9.08	1.31	9.59	0.80	9.86	0.38	9.00	1.09
Q4	8.17	2.17	8.17	0.87	9.57	1.13	9.18	1.25
Q5	8.83	1.80	9.82	0.39	10.00	0.00	9.29	1.08

**Table 3 sensors-19-02952-t003:** Average percentage of progress of the students in each lab activity. Activities are shown in the table in chronological order of appearance in the course material from left to right.

Lab Activity	Average Progress (%)
PL_EC2	72.44
PL_EC2_S3	52.73
PL_RDS	43.59
PL_APP	40.16
PL_CF	36.09
PL_VPC	32.51
PL_SERVERLESS	16.56

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
