# Peer review of "A Visual Dashboard to Track Learning Analytics for Educational Cloud Computing"

_sensors, 2019, doi:10.3390/s19132952_

Reviewer 1 Report

The purpose of this study is to introduce CloudTrail-Traker which is an open-source serverless platform and supports instructors to track students' engagement and achievement. I have several concerns to recommend the publication of this paper:

There are somehow many jargon which make the reader understand the intentions of this study. For example, AWS, CloudTrail-trakers, CloudTrail logs, LASyM, Amazon S3, FaaS, REST API, etc. Although there was additional explanation and annotation, such terms were hard to follow the writing. 

The paper could be improved if the paragraphs are densely organized.  For example, in the introduction (p. 1 line 23-27), there is a long sentence which becomes a paragrph. Normally, a paragraph consists of three to four (or four to five) sentences that has a main idea.

I think this study is meaningful in terms of that the study attempted to utilize a method to develop the visual dashboard for teachers with the low(free) cost. However, if the study highlights the strengths and weakness of using this technology(e.g., AWS) more clearly, it would be better.

The title of this paper is "A Visual Dashboard~". And in the conclusion, the authors expressed like this: "... having a high-level graphical user interface that helps them know wheter they properly termincated the allocated computing and storage resources reinforces this skill." I agree with this. However, overall, the paper did not highlight the aspect of visual dashboard". That is  what kind of visual strategies were applied this dashboard was missing. 

Finally, this study introduces a case study: usage in Cloud Computing Education. And the case study only reports the result of survey data in using a questionnair including five questions.  This is a weakness of introducing one case study as an empirical study. If the paper introduces two to three cases and compre the results or reports both quantitative analysis and qualitative analysis, the credibility of the study would be improved.

Author Response

“Thank you for the review. Please find attached our memo of changes in order to improve the paper”.

Reviewer 2 Report

The paper describes a learning analytics dashboard used for courses that use the AWS Cloud services.  

While the topic of dashboards, especially those oriented to particular fields, is worth pursuing, I have several issues with this paper.  I will try to summarize them in the following points:

- There is no real evaluation: A survey is not a good way to evaluate dashboards.  The overwhelmingly positive response is most probably related to bias than to real impact of the dashboard.  If you want to evaluate the dashboard, a proper experiment should be used (with control groups) and the change on the behavior of students or instructors should be the variable to measure.  

- Lack of theoretical background: While this is not an educational journal, the subject being treated deal with education.  As such, there is no description of how the design of the dashboard was guided by educational theory and that what is presented is believed to affect students understanding of their own behavior.  If you design a sensor, that design should be based on the physics theories that rule the process being measured.  In this case, the psychology of the students/instructors is what should govern the design decisions.

- Impact in student behavior: The tool seems to be a great way to monitor the use of the AWS services, however, I am not completely sure it has a real impact on "understanding and optimizing" learning.  Having a progress check indeed is useful for self-regulation, but it is not clear how students/instructors can convert this information into actions.

In summary, I think this is a useful tool with several technical novelties, but it is not what I would consider a research paper nor for the sensor community or the learning analytics community. 

Author Response

(The authors gave the same response as above.)

Reviewer 3 Report

GENERAL

Congrats for your work. You have developed an original digital artifact and used it in different subjects, successfully... Said that, your work is interesting, but needs to address the following points:

ABSTRACT

- Be more specific. For instance, the phrase "However, the wide panoply of services involved, together with the pay-per-use model of a public Cloud, imposes serious challenges."... which is the more important challenge? It is important to set the context to introduce the problem.

- Set the problem. Why the teachers require tracking of students and an overview of the resources? Or the problem is that students need help with their self-regulated learning process for any reason? Explain what aims to solve your work.

- The abstract should be written in order to describe the context, the problem, the main objectives, methodology, and results. Please, rewrite your abstract to present those elements.

METHODOLOGY & RESULTS

- Your paper will be more clear if you rearrange your text to add two specific sections in order to introduce the "Methodology" used (literature review, satisfaction survey methodology&questions, development/visualizations considerations in regarding literature review...) and "Results". 

CONCLUSION

- Support conclusions with the results.

- If you have empirical results about the impact when students use the dashboard (instead of the satisfaction survey) expose them.

Author Response

(The authors gave the same response as above.)

Reviewer 4 Report

Great work on the Cloud Tracker and sharing your work. This is an important topic especially since you are working toward making it easier for learners to see their progression. 

My one concern is that several of the statements imply a strong connection between the platform and learning processes. For example, you state (lines 315-16) "disclosing this data to the students themselves is bound to have an impact on self-regulation, where students become aware of the activities already carried out but, most important, those that are still pending." This is a good way to support self-regulation but whether this impacts self-regulation is an unknown given that who the learners are is an important aspect of how they approach the use of the tool and their learning. Please review the paper with a critical eye in relation to the claims of impact on learning and consider softening your language unless you are referencing prior research. 

Author Response

(The authors gave the same response as above.)

Reviewer 5 Report

The paper presents a proposal of CloudTrail-Tracker, a platform to track students’ activities in shared AWS accounts for learning cloud computing.

Although the authors mention related work and other tools that offer similar features, there is very little detailing about what they do and how they are related to the study presented in the paper.

On Section 4.1, there is very little detail about the sample of participants who evaluated the system.  The authors jumped straight to the results, without better explaining the instrument used in the evaluations, the profile of the participants, and how the study was conducted.  This way, it makes it very difficult to understand the details and to analyze the conclusions.

This section also provides a couple of very brief comments from some of the participants.  I would strongly suggest the authors perform a more in-depth discussion of these results and on the implications for the design of the system.

I would also suggest the authors provide a more in-depth explanation of the types of analyses enabled by the system they implemented, in terms of the use by students and lecturers.

Regarding the suitability to the journal, the authors have mentioned only one single time that they consider the data collection for the analytics as “virtual sensors”.  It is very important that this concept be better explained in the methods, results, and in the conclusions of the study.

In general, the paper is well written.  However, I recommend the authors proof-read the paper and correct a couple of issues.  Following I present a non-exhaustive list of issues that I have encountered.

Citations are metadata, and should not be part of the flux of text, such as in: “Therefore, as described in [6]”.

Page 2, line 56 - take advantage of _these_ data

Page 3, line 107 - Amazon Web Services _offer_ two

Figure 1 is shown on the paper before being mentioned in the text

Page 5, line 182 - -use features, _it_ (?) has a very

Table 1 also appears before its first mention

Author Response

“Thank you for the review. Please find attached our memo of changes in order to improve the paper”.

Round  2

Reviewer 1 Report

I think the issues that I mentioned in the first review were well solved throughout the revision.

Author Response

Thank you for the suggestion improvements.

Reviewer 2 Report

Thanks for the changes made to the manuscript.  However, I think that most of my initial issues with the paper are still present in this new version:

- No real evaluation: Even if an experiment with a control group is not possible, at least you could gather data about the usage of the dashboard (how many times was it visited, the visits increased or decreased with familiarity, is there any correlation between time of visit and actions taken by the student.).  In its current form, the evaluation of the dashboard is subpar to any of the reputable published literature on the subject.

- Lack of theory: While you mention that there is no "golden rule" to design dashboards, there are at least educational motivations on why given information is presented.  Why you present the information that you do in the way that you do it?  Is there any justification that can be extracted from the educational theory?

Again, I know that your system is indeed technically correct and could be useful for the students, but its reporting is far from the standards in my field (educational technology).

Author Response

Thank you for the review. Please find attached our memo of changes in order to improve the paper”.

Reviewer 5 Report

The authors have performed a detailed revision of the study, following the suggestions given in the previous review round.

In particular, the authors have included mentions to the "virtual sensors" in other parts of the paper.  However, as this would be one of the main topics that link the paper with the journal, I feel that it would be important to provide a more detailed explanations of what the "virtual sensors" are, and what characterizes them as sensors.

There is a more detailed description of related studies in the "related work" section.  However, they are all in one paragraph, which makes reading difficult.

To some extent, the authors have discussed their results in comparison to related work and the implications of the results.  However, I still feel that the discussion could be done with more depth and including other studies that were mentioned in the discussion.

Author Response

(The authors gave the same response as above.)
